

**Communication strategies to address geo-hydrological risks:**
**the POLARIS web initiative in Italy**
P. Salvati[1], U. Pernice[2], C. Bianchi[1], I. Marchesini[1], F. Fiorucci[1], F. Guzzetti[1]
(1) Consiglio Nazionale delle Ricerche, Istituto di Ricerca per la Protezione Idrogeologica, via Madonna Alta
126, I-06128 Perugia, Italy
(2) Innovation Consultant, Viale Michelangelo 2315, I-90135 Palermo, Italy
*Correspondence to:* Paola.Salvati@irpi.cnr.it, Tel. +39 075 50144427, Fax +39 075 5104420



**Abstract**. Inundations and landslides are common phenomena that cause serious damage and pose a severe
threat to the population of Italy. The societal and economic impact of landslides and floods in Italy is
particularly severe, and strategies that target the mitigation of the effects of these events are essential.
Although, in the last few years, the scientific community has wanted to communicate information on research
activities regarding geo-hydrological hazards and the associated risks to society through thematic websites,
very often, communication achieves specific technical purposes for experts. To address the problem posed by
the lack of communication on geo-hydrological hazards with potential human consequences in Italy to the
broader society, we designed the POLARIS website. The POLARIS website publishes accurate and detailed
information on geo-hydrological risks, including periodical reports on landslide and flood risk to the
population of Italy, data and analysis on specific damaging events and blog posts on landslide and flood
events that able to encourage mass media and citizens' engagement. By monitoring the access of users to
POLARIS since January 2014, when the website was published, we registered maximum access during the
occurrence of the worst geo-hydrological events and for the promotion of relevant new content through press
releases. In particular, in the latter case, we noted the highest access value when journalists promoted the
website through television channels. The POLARIS initiative demonstrates how the scientific community can
implement suitable communication strategies that address different societal audiences by exploiting the role
of mass media and social media. These strategies can help these audiences to understand how risks can be
reduced through appropriate measures and behaviors; thus, they can contribute to increasing the resilience of
the population to geo-hydrological events.




## 1    Introduction

Geo-hydrological hazards, such as inundations and landslides, are common phenomena that cause serious damage and pose severe threats to the population worldwide. Currently, river flooding annually affects 21 million people worldwide; this estimate could double to 54 million people by 2030 (www.wri.org). For landslides, Petley (2012) demonstrated that losses due to landslides were considerably higher than had been previously considered. Global costs of geo-hydrological disasters have increased in recent decades and, in future decades, it is expected that the number of people at risk and the occurrence of extreme events will both grow (https://www.ipcc.ch). Integrated risk management involving public authorities, researchers, companies and citizens is required to address the interconnectivity between physical infrastructures, economic systems and the role of human factors (Jonkman and Dawson, 2012). Therefore, developing effective risk communication strategies as an integral part of risk management must be a priority for risk managers and regulators (Frewer, 2004). Education and the communication of geo-hydrological risks enable more effective decisions and knowledge-based actions between decision makers, land planners, experts and the affected population; in addition, non-structural mitigation measures could become helpful tools to develop a more resilient society (Höppner et al., 2010).

Knowledge-oriented risk communication campaigns on the causes and dynamics of geo-hydrological hazards and their possible consequences to human life, conducted with relevant frequency, can effectively increase public awareness of these hazards. The appropriate transfer of knowledge between experts and the public can be facilitated using appropriate communication strategies and programs at the national level to align the views of the public with those of experts (Frewer, 2004). O'Neill (2004) explains how the effectiveness of risk communication depends on multiple factors, including a complex interaction between the characteristics of the audience, the type and content of the message and the characteristics of the medium. Considering such a composite interaction, communication can foster effective plans and responses to geo-hydrological risks by fostering the capacity of a broad range of targets (i.e., individuals, groups and organizations) to prepare, manage and recover from geo-hydrological events. In addition to developing knowledge, many risk communication efforts are made to change people's attitudes towards the types of hazards that they may encounter.

Thus, the availability, at a national scale, of detailed and organized information on the geographical and temporal distribution of geo-hydrological events and their consequences is fundamental to implement national education and preparedness programs.

In Italy, detailed information on landslides and floods is available, and catalogues of landslide and flood events with fatalities have been organized and constantly updated (Guzzetti et al., 1994, 2005; Guzzetti and Tonelli, 2004; Salvati et al., 2010, 2012, 2013). For this country, in recent decades, much effort has been



exerted to analyze landslide and flood hazards and the associated risk at various geographical scales, from the
site specific (local) to the synoptic (national) scale. Despite these efforts, most of this study remained
unknown to the public, who ignore the possible damaging effects that landslides and floods could produce
(Salvati et al., 2014). Despite the large number and wide geographical distribution of landslide and flood
events, the Italian population receives minimal information and has minimal knowledge on the type,
characteristics, frequency and severity of the harmful events that have occurred in the territory where they
live. This finding is confirmed by two national surveys conducted in 2012 and 2013 through
Computer-Assisted Telephone Interviews (CATI), to measure the perception of landslide and flood risk by
Italy's population. The low risk perception demonstrated by the Italian population reflects a lack of
knowledge and a weak motivation to learn. This result is surprising because accurate and timely information
is fundamental for the implementation of risk mitigation and adaptation strategies (Salvati et al., 2014).
Although, in the last few years, the Italian scientific community has begun to communicate information on
geo-hydrological hazards and the associated risks through communication initiatives and thematic websites
(http://avi.gndci.cnr.it/;                                                    http://sici.irpi.cnr.it/;
http://www.isprambiente.gov.it/it/progetti/suolo-e-territorio-1/iffi-inventario-dei-fenomeni-franosi-in-italia;
http://www.pcn.minambiente.it/GN/), these very often suffer from a lack of appropriate communication
strategies that address the various targets with the most suitable media. Consequently, these initiatives remain
mainly addressed to experts for specific technical purposes with content and web interfaces that are barely
appreciated by a wider audience and rarely synchronized with social networks. Various problems emerged
when designing a communication strategy. First, public interest in the issue is important. As Keys (1999)
noted, "It has been apparent for some time that creating community awareness of floods and storms is not
easy, (….) Most of the time, people are not particularly interested in them" (O'Neil, 2004). The core of the
problem is to capture the attention of the public and, with long-term actions, habituate people to be interested
in the topic. Second, it is important to find the appropriate mediators to reach the largest number of people.
Media can represent key mediators of communication between different audiences, i.e., the public, the
scientists, the policy-makers and the operational management (Beck, 1992).
The mission of the POLARIS website is to provide correct and reliable information mainly to media, which
will help to further communicate such information to other audiences. In addition, the role of social media
should be carefully considered to engage types of audiences that are usually weakly interested in information
regarding geo-hydrological risk. Thus, efforts were made to improve the link between the POLARIS website
and the Facebook page (https://www.facebook.com/CNR.IRPI) of the Istituto di Ricerca per la Protezione
Idrogeologica (IRPI, www.irpi.cnr.it), of the Italian Consiglio Nazionale delle Ricerche (CNR,
www.irpi.cnr.it), by conveying immediate and concise information on natural disasters (also through pictures
and videos), interspersed with invitations to visit the POLARIS website for detailed information.
In this paper, we begin by explaining the website information architecture; we analyze the users' navigation



data during the 21 months since the website was published. Then, we explain possible relations between the
maximum access and the contests in which they occurred. Finally, we conclude by summarizing our findings.
**2  Nomenclature**
In this work, we adopted the terminology and the definition available from Google Analytics. We use the term
*session* to mean the period of time a user is actively engaged with the website or an app. All usage data
(screen views, events and ecommerce) is associated with a session. The *users* are people who have had at
least one session within the selected date range; this includes both new and returning users. The *pageviews* are
the total number of pages viewed, including repeated views of a single page. The *source* is the place users are
before viewing the website content, such as a search engine or another website. *Referral traffic* is Google's
method of reporting visitors that arrived at a specific site from sources outside its search engine.
**3  POLARIS website**
The effectiveness of the POLARIS communication strategy relies on the main assumption that the scientific
community can play a key role in increasing the awareness of individuals and groups on geo-hydrological
hazards and on the type and extent of the risk posed by geo-hydrological hazards to the population. This role
should be achieved working in two directions: (i) providing Mass Media (such as journalists) with correct and
reliable information, which they can further communicate to the broader civil society, and (ii) adopting less
technical and more widely comprehensible language to better engage citizens. Figure 1 shows the
communication flow in POLARIS, where the scientists use different communication approaches to Mass
Media, Civil Protection and Local-Regional Authorities, and to Citizens. The Media sector captures
information from scientists and uses it for communication purposes.
The scientific and technical content is based on communication strategy that avoids scientific and technical
terminology towards a more widely understandable language. For this purpose, consultants experienced in
web-communication strategies on natural hazards, info-graphics and user experience design were involved in
the initiative. The consultants' contribution consists in arranging the messages using intuitive and engaging
web interfaces to display data, graphs, tables, video and carefully considering usability and accessibility of the
website to diversified audiences.
POLARIS is based on a well-defined information architecture encompassing six main sections: (i) Reports,
(ii) Are you ready?, (iii) Events, (iv) Alert Zones, (v) Focus and (vi) Blog. These sections provide different
and complementary information, respectively, which include (i) periodical reports on landslide and flood risk
to the population of Italy, (ii) suitable behaviours to adopt during damaging events, (iii) data and synthetic
analyses on specific disasters, (iv) visual information on the geology, the geomorphology and on the
damaging events of the Italian Alert Zones defined by the National and regional Civil Protection Authority for




geo-hydrological hazards, (v) in-depth analysis of relevant topics or extraordinary events that particularly
damaged the population and (vi) blog-posts on landslide and flood events encouraging citizens' engagement.
Figure 2 shows the POLARIS home page we structured with images and illustrations as helpful tools to
browse the site.

### 3.1 POLARIS website structure

The "Reports" section illustrates periodical reports on landslide and flood risk to the population of Italy.
Reports are published every six months. The last report is available in two formats: an on-line version, and a
PDF format. The on-line report is directly integrated with the CNR IRPI Spatial Data Infrastructure, SDI
(Salvati et al., 2013) in which the database is located and has access to data that is always updated. Each
report contains the list of landslides and floods that occurred in the period (six months or a year), with
information regarding date, location, deaths (in Italian: morti), missing persons (in Italian: dispersi) and
injured people (in Italian: feriti), maps, statistics and an analysis of the landslide and flood events that caused
direct consequences to the population. Statistics are available for different periods of one, five and fifty years,
which enables comparative analyses of the geographical and temporal variations of geo-hydrological risks in
Italy.
The "Events" section publishes information on specific meteo-climatic events that have occurred in Italy
using text, maps, videos, photographs and drawings. In this section, specific icons were designed to define the
type of geo-hydrological events. In addition, a short description containing information on localities affected,
damages and victims is provided with a map of the sites affected by the landslides and floods that affected the
population.
The "Focus" section publishes information on specific topics, provides analysis for each Italian region and
provides explanations of single historical or recent disastrous events.
Both the "Events" and "Focus" sections inform the population on the extent and severity of geo-hydrological
risks; in addition, they are important sources of data for Mass Media, which can sensitize larger number of
citizens.
The "Alert zones" section shows the 134 Alert Zones defined by the Italian National and regional Civil
Protection Authorities to help better forecast when and where geo-hydrological hazards (including landslides
and inundations) can occur and their impact. This section provides the possibility to query a number of
information items, and a sidebar offers access to a set of layers and maps (i.e., lithological and morphological)
for each Alert Zone.
The "Are you ready?" section contains information on suitable behaviours to adopt before, during and after a
damaging flood event; this section provides elementary behavioural rules that may save people's lives.
Finally, the "Blog" section encourages bottom-up participation by users who can post comments on



geo-hydrologic hazards and risks.
On the home page, particular focus is reserved for "Happened Today" (Italian: *Accadde oggi*), which is a
daily register of events in which, for each day of the year, POLARIS publishes a short description of those
relevant events that adversely impacted the population that specific day. This section is directly linked to the
CNR IRPI SDI, which daily automatically relates the event to the exact day.
**4    Data**
We used Google Analytics to monitor the traffic and performance of POLARIS. In particular, we focused our
analysis on (i) channels used, (ii) number of sessions, (iii) number of users, (iv) users viewing of the entire
website and its single pages and (v) the geographical distribution of access.
We also monitored POLARIS' Facebook page through the "Insight" instrument and, in particular, the number
of "likes" expressed by users or the number of users who viewed the posts. Moreover, we performed an
analysis of the types of posts (containing, video, link, images or text alone) that primarily interested users and
their provenience.
**5    Analysis and results**
In this section, we describe the analysis we performed to identify possible trends of interest to the POLARIS
content and the relation between the peak access values to the website and the possible causes that increased
the public interest in the website content. We also performed similar analysis for the CNR IRPI Facebook
page, which is the Institute's most active social network.
**5.1    POLARIS website**
The analysis of the data series available from Google Analytics for the period of the website publication, from
16 January 2014 to 15 October 2015, allows the elaboration of certain general statistics summarized in Table
1, in which we listed the data separately for sessions, users, pageviews and referrals from social networks. We
studied the geographical distribution of the website users and the number of pageviews for each website
section. The results are shown in Figure 3.
Because POLARIS is published in Italian, it is not surprising that the sessions mainly originate from Italy
(91%). Figure 3a shows the geographical distribution of the sessions in Italy. The small percentage of sessions
originating from other nations is concentrated in the USA, China, Japan and Germany. Darker and larger dots
in the map show the increasing number of sessions, with few areas in which sessions are highly concentrated.
The largest number of sessions originate from the Umbria Region, where the main office of our institute is
located. Other areas where POLARIS was frequently accessed were Rome, where the majority of the




government offices are located, Milan (Lombardy region), Turin (Piedmont region), Genoa (Liguria region)
and Palermo (Sicily region). These cities host institutes and researchers who are interested in
geo-hydrological issues. The pie chart on the right of Figure 3 (b) reports the number of pageviews for the
different sections of the website. The home page is the most viewed page, containing, in addition to the
navigation menu, the Happened Today (*Accadde Oggi*) section, which is read by many people because its
content changes daily. The second most viewed section is the Report section, which publishes periodic reports
on the risks posed to the Italian population by landslides and floods; this is updated every six months and
allows the download of reports. The Focus and the Event sections have similar access percentages; their
content can be very easily read and is accompanied by explicative figures and maps. The content differs in the
subjects; on the Focus page, we discuss in-depth issues related to geo-hydrological hazards and risks, whereas
the Events section is dedicated to the description of specific events that caused damages to the Italian
population. The Alerts Zones and the Are You Ready? sections were not accessed as much as we expected,
although they both contain relevant information and advice to help develop more suitable behaviours toward
disaster resilience.
Monitoring the number of sessions during the 21 months since the website's publication, it was possible to
elaborate their temporal distribution. For the purpose, the number of sessions per day was normalized to the
daily average number of sessions in the 21-month period (long-term average, 26.9). The results are shown in
Figure 4, where the ratio in the x-axis represents the daily access number divided by the average access
number during the observation period. The grey parts of the line show periods below the long-term average,
and the blue parts show periods above the long-term average. An inspection of Figure 4 reveals that there was
an increase in the number of sessions (blue dashed line in Fig. 4); however, a significant variation in the daily
distribution is also apparent. It is observed that, in 350 days of 2014, 42 days (12%) were above average and
308 days (88%) were under average. In the 288 days of 2015 (until 15 October 2015), the trend changed, and
there were 182 days above average, which corresponds to the majority (63.2%) (Table 1).
To find a possible repeating pattern or periodic signal, the time series data related to the number of sessions
were analyzed using the autocorrelation function (ACF). The ACF measures the degree of correlation
between a signal and the signal itself shifted by a given lag, and is defined as:
$$ACF = \frac{1}{n\sigma^2} \sum_1^{n-k}(X_i - \bar{x})(X_{i+k} - \bar{x}) \qquad\qquad\qquad \text{eq. (1)}$$
where $k$ is the lag (a day in this work), $n$ is the length of the time series (607 days), $\sigma$ is the standard deviation
of the values (i.e., the standard deviation of the number of sessions), $\bar{x}$ is the average of the values (i.e., the
average of the number of sessions), and $X_i$ is a given value of the time series (the value of the number of
sessions of the day $i$). Due to the evident increasing trend (non-stationary) in the average number of sessions
during the observation period (dashed line in Fig. 4), the ACF has been estimated with a kernel smoother that
uses a normal weight function to average nearby observations in a bandwidth of 100 days (Fig. 5a).



The plot of Figure 5b shows the coefficients (ACF) calculated per different lag times. The autocorrelation
value can vary between 1 and -1, and the area between the blue dashed lines represents non-significant
autocorrelation values. The analysis showed that the value of ACF decreases when the lag $k$ (days) increases
and that a marginally significant value of autocorrelation can be observed only for a lag of seven days.
However, because the correlation value is not significant at 14 or 21 days, we conclude that the time series of
the number of sessions of the POLARIS website do not present evidence of a periodic pattern. The same
analysis was performed using the residuals with respect to the linear interpolation of the data (dashed line in
Figure 4). Again, the analysis does not provide evidence of periodic signals.
To gain a better understanding of the temporal distribution of the access and to identify the peak values, we
used the daily number of users and pageviews from Google Analytics. We then related the peak values with
several factors, including (i) the occurrences of harmful events, (ii) the daily early warnings from the Italian
National Department of Civil Protection, (iii) the publication of new content, (iv) the press releases published
with our data and (v) the promotion of the website through media.
Figure 6 reports the daily statistics of users (Fig. 6a) and a comparison between the users and the number of
pageviews (Fig. 6b) for the 21 months of website publication, with icons positioned to visualize the possible
relations. We note how the relation between the peak values and the occurrences of the harmful events until
December 2014 became increasingly less relevant since the early months of 2015. In particular, during the
period ranging from 15 January to 31 December 2014, the majority of the peaks were registered in the interim
of the harmful event occurrences, i.e., on 16-22 January (25 users, 51 sessions and 425 pageviews), when the
two Italian regions of Liguria and Emilia Romagna were hit by heavy rain, which caused two fatalities, and a
railway interruption to France was caused by a landslide. Similarly, on 6-15 October, an event hit Liguria and
other regions in the North of Italy causing four deaths and generating a peak with 44 users, 48 sessions and
115 pageviews. Other correspondences were identified with the icons used to indicate the events, the same as
those we used to indicate the type of event (landslide, flood and geo-hydrological events) on the website.
Other peak values were related to the publication of new contents. A peak occurred on 15 September 2014
due to a post dedicated to a relevant paper published by the CNR IRPI researchers (38 users, 50 sessions and
110 pageviews); it also occurred on 19 November, due to the publication of the "Are you ready?" section,
explaining how to behave during geo-hydrological events (80 users, 94 sessions and 192 pageviews). The
maximum value was registered when the website was promoted through television by a meteorologist during
an evening national broadcasting program (338 users, 362 sessions and 951 pageviews).
Another important value corresponds to the press release launch on 13 January 2015, to disseminate the
annual report on the geo-hydrological risk to the population; this was prepared for 2014 and available in the
Report section (119 users, 141 sessions and 436 pageviews). After these announcements, the site has begun to
be consulted by journalists and technicians of different government offices and agencies working on land
management. This finding is confirmed by the publication of POLARIS's maps and statistics in national


newspapers and in on-line media corresponding to major event occurrences that captured the interest of the
public and to the citation of the website URL in reports published by national or regional institutions. The
finding means that POLARIS offers quick and easy access to essential information on geo-hydrological
hazards and risks.
During 2015, the relation with the occurrence of the events decreased; however, the relation with the
publications of new content became more significant. Analyzing the sources where the POLARIS traffic
originates daily, we found that other peaks were the consequences of the daily activity of users from
government offices or agencies. In Figure 6b, we plotted the users and the pageviews data together. The mean
number of pages per user, in the entire period, was 2.5; however, the inspection of Figure 6b reveals that the
variability of this ratio is very large, and days exist when the mean value has been largely exceeded. This
result demonstrates that people browse through the site's pages before leaving.
We maintain that the relation to the occurrences of harmful events depends on the new, specific content and
the videos that are published during or immediately after harmful events not only on POLARIS but also on
the CNR IRPI social network pages from which people can directly access POLARIS.

## 5.2   CNR IRPI Facebook

Each new content item published on POLARIS was shared via Facebook and Twitter, the two most popular
social networks in Italy. We use Facebook and Twitter CNR IRPI accounts to disseminate simple and
immediate messages addressing the geo-hydrological hazards. In particular, the objective is to increase the
public awareness of the frequency and proximity of the geo-hydrological events and to disseminate media
showing hazardous behaviors that pose serious, fatal risks to people.
Analyzing the number of referrals from the social networks, corresponding to 14% of the total, we found that
the majority (80%) originates from Facebook. The simpler modality of sharing content offered by Facebook
with respect to a website makes the publication of links and videos easier. Social media is very widely used
when a severe weather condition is occurring. Therefore, it is relevant to compare the number of people who
have viewed the content of the CNR IRPI Facebook page with the occurrence of extreme rainfall conditions
and or severe warning declarations of the Italian National Department of Civil Protection. For the purpose, we
used Facebook statistics because it is the social network from which the majority of the access to POLARIS
was registered.
To define the extreme rainfall conditions that occurred in Italy, we exploited an analysis based on hourly
rainfall measurements. The analysis was performed in the 84-day period between 1 August and 23 October
2015. We exploited sub-hourly rainfall measurements by more than 2000 rain gauges distributed over the
entire Italian territory. According to the method described by Rossi et al. (2015), the empirical cumulative
distribution function (ECDF) of the cumulative rainfalls has been modelled for each rain gauge. The function




allows the calculation of the non-exceedance probability for any given cumulative rainfall and for a set of
predefined durations (3, 6, 12, 24 h), which estimates the non-exceedance probability of the cumulated
rainfalls, for each rain gauge. To obtain a continuous representation for the entire Italian territory, the rain
gauge data have been interpolated using an inverse distance weighted (IDW) algorithm. This process resulted
in a set of four (one for each duration) raster maps that show the non-exceedance probability of the
cumulative rainfalls. The maps have been analyzed to identify the days when at least 10% of the Italian
territory has been interested by a non-exceedance probability of 80%. This probability value corresponds to
cumulative rainfall events that can be defined as extreme events and that could have triggered
geo-hydrological events.
The results of the analyses showed that, in the considered period, the extreme conditions occurred six times
for a duration of 3 h, 12 times for a duration of 6 h, 15 times for a duration of 12 h, and seven times for a
duration of 24 h. We plotted these extreme conditions in the daily distribution of Facebook users shown in
Figure 7. We observed that extreme conditions, represented by blue dots on the basis of their duration,
occurred on 16 days (19% of the days in the investigated period), grouped into 11 meteorological events that
lasted one or more days. In the graph, we plotted with a red icon the days for which it is known that severe
warnings of the Italian National Department of Civil Protection were enacted; the days when severe
geo-hydrological events occurred are shown in orange in Figure 7. Analyzing the four highest peaks, the first
(September 16, 2060 peak value) corresponds to the publication of videos and images regarding the
Piacentino (Emilia-Romagna region) flood event of September 13-14, 2015, which caused three deaths and
serious damage. The second event on October 6 corresponds to the publication of a re-visit of the Vajont
disaster (the most disastrous landslide event that has occurred in Italy) in POLARIS at a date near the event's
anniversary; this was immediately shared with Facebook. A few days later, on October 10, the publication of
a video showing cars dragged by the water flow caused by heavy rainfall in the Tyrrhenian Messina area
(Sicily region) caused a 3916 peak value; finally, the peak of October 21 related to the publication of content
that triggered a strong debate. Although the 3-month investigation period is very short, we can observe that,
apart the first half of August, there is suitable correspondence with the rainfall extreme conditions and the
peak values of Facebook access. In addition, the peak values correspond to the content published and that
people shared.
**6    Concluding remarks**
The analysis we conducted in the 21 months after publication of the POLARIS website allowed the following
considerations. The geographical distribution of people interested in the published topics is widespread
throughout Italy, with a few geographical areas in which sessions are highly concentrated. After the home
page, the most viewed website section is the Report, followed by the Focus and Events sections. In a period
shorter than two years, the number of sessions has generally increased; however, we observed that, in 2015,



the most significant positive step occurred. The analysis of the time series, performed to identify possible
periodical signals in the daily distribution of sessions, did not highlight any relevant information.
Monitoring the access of users to the POLARIS website and the number of pageviews during its publication
period from 16 January 2014 to15 October 2015, we noticed that, frequently, the peak values correspond to
the occurrence of particularly damaging geo-hydrological events. However, inspection of the daily statistics
available for CNR IRPI Facebook demonstrated that a correspondence exists between the extreme rainfall
events and the number of people who have viewed the content Facebook page. This finding was expected
because CNR IRPI Facebook page's objective is to capture the attention of the public at large by proposing
content that satisfies their curiosity and their immediate interest during extreme events, which increases the
number of followers. Because the Facebook page is linked to POLARIS, an increase in Facebook followers
can trigger a gradual increase in the number of people interested in more structured and specialized content
and data on geo-hydrological topics such as those published on POLARIS. Similarly the specificity,
scientifically based, of the POLARIS content, which is focused on geo-hydrological hazard and risk, became
a source of information for journalists and media operators. The growth of user access when media operators
publicized the website, suggested that we enhance our collaboration with scientific journalists by linking
traditional (e.g., television) and social media to further enlarge the awareness of the website, and to better
explain to users how to exploit the website information.
The POLARIS initiative demonstrates how the scientific community can implement different communication
strategies to enhance an effective process that helps different audiences to understand (i) how risks associated
with geo-hydrological hazards are estimated and (ii) how risks can be reduced by increasing knowledge to the
population.
**Acknowledgments**
We thank Salvatore Buda and Vito Lo Re for the website design and info-graphic development, and Mauro
Rossi for making available the non-exceedance probability cumulative rainfall maps' rainfall data series.

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



**Figure captions**
Figure 1. The POLARIS communication flow.
Figure 2. The POLARIS Home Page (http://polaris.irpi.cnr.it). Violet box is the English translation from
Italian.
Figure 3. General statistics from Google Analytics for the 638 day period from 16 January 2014 to 15 October
2015. The map on the left (a) shows the sessions' geographical distribution for the Italian territory. The pie
chart on the right (b) reports the number of pageviews for different sections of the website.
Figure 4. Daily average access number to POLARIS in the 16 January 2014 to 15 October 2015 period.
Figure 5. The plot on the left (a) shows the autocorrelation coefficient calculated using the time series of the
number of sessions of the POLARIS website. The plot on the right (b) shows the kernel smoother using a
normal weight function.
Figure 6. Daily statistics of the number of users (a) and a comparison between the pageviews and the number
of users (b) for the 21 months of site activity in the 16 January 2014 – 15 October 2015 period.
Figure 7. The plot shows the number of unique Facebook page users. In the plot, the days when the extreme
rainfall condition occurred were reported with blue dots; the major geo-hydrological event occurrences were
reported in orange, and the severe warning declarations were reported with red dots.





Table 1. The Table lists the POLARIS website general statistic for sessions, users, pageviews and referrals
from social networks, calculated using Google Analytics data.

| | POLARIS Statistics | Number |
|---|---|---|
| Sessions | Total | 17,159 |
| | Daily average | 26.9 |
| | Average duration | 00:02:38 |
| | Days above average (2014) | 42 (12%) |
| | Days above average (2015) | 182 (63.2%) |
| Users | Total | 11,529 |
| | Daily average | 23.3 |
| | Days above average (2014) | 37 (10.6%) |
| | Days above average (2015) | 180 (62.5%) |
| Pageviews | Total | 44,032 |
| | Daily average | 69 |
| | Average per session | 2.6 |
| | Days above average (2014) | 68 (19.4%) |
| | Days above average (2015) | 165 (57.3%) |
| | Home page | 14,284 |
| | Report section | 5976 |
| | Focus section | 5509 |
| | Significant Event section | 5489 |
| | Blog section | 2550 |
| | Alert Zones section | 2108 |
| | Are You Ready? section | 1894 |
| Referrals | Total from Social Network | 2394 |
| | Facebook | 1917 (80%) |
| | Twitter | 430 (18%) |
| | Other Social Networks | 47 (2%) |





414 Figure 1. The POLARIS communication flow.


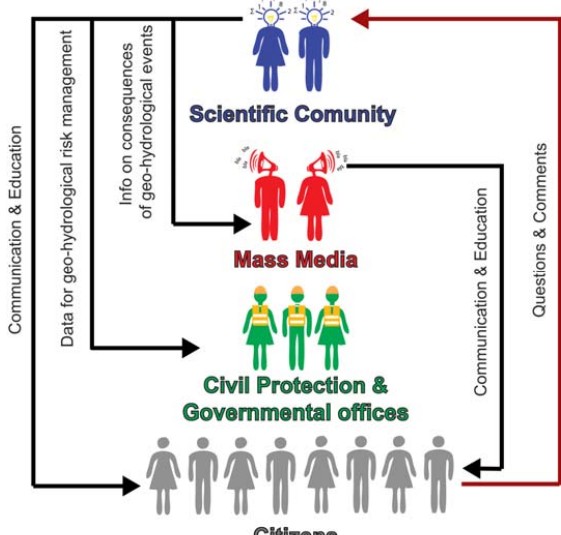







Figure 2. The POLARIS Home Page (http://polaris.irpi.cnr.it). Violet box is the English translation from
Italian.

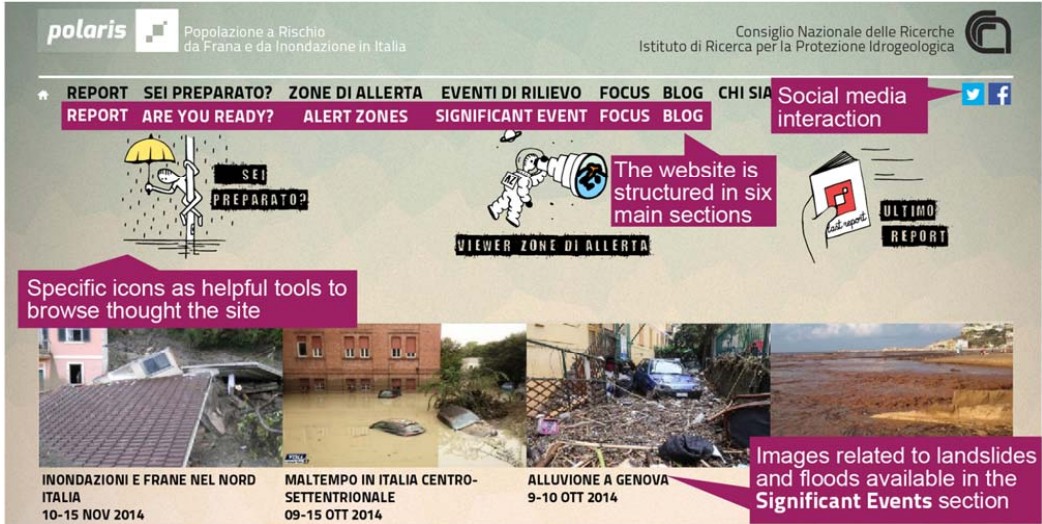






Figure 3. General statistics from Google Analytics for the 638 day period from 16 January 2014 to 15 October
2015. The map on the left (a) shows the sessions' geographical distribution for the Italian territory. The pie
chart on the right (b) reports the number of pageviews for different sections of the website.

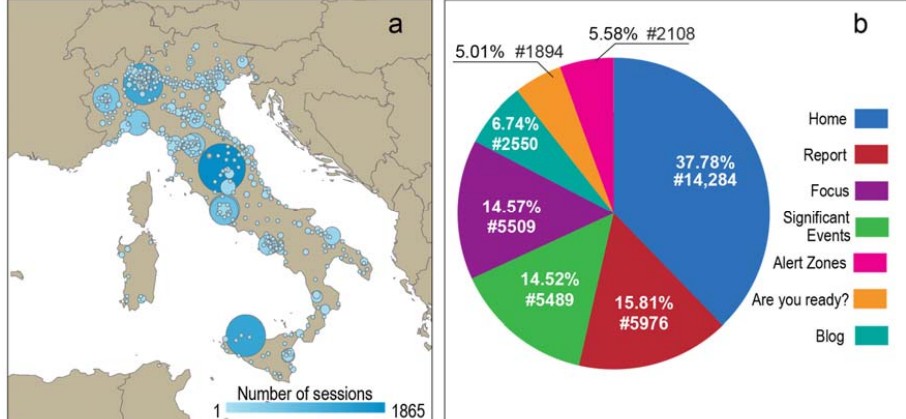






Figure 4. Daily average access number to POLARIS in the 16 January 2014 to 15 October 2015 period.

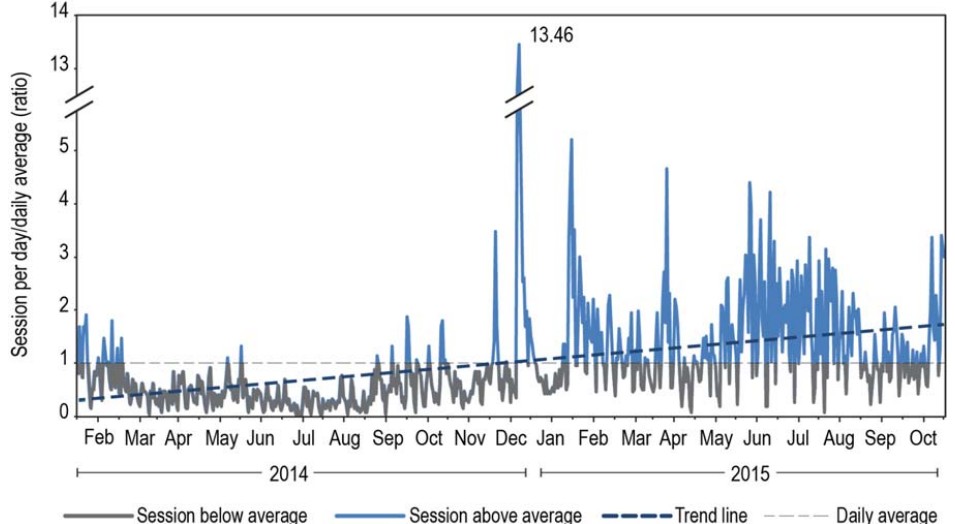







Figure 5. The plot on the left (a) shows the autocorrelation coefficient calculated using the time series of the
number of sessions of the POLARIS website. The plot on the right (b) shows the kernel smoother using a
normal weight function.

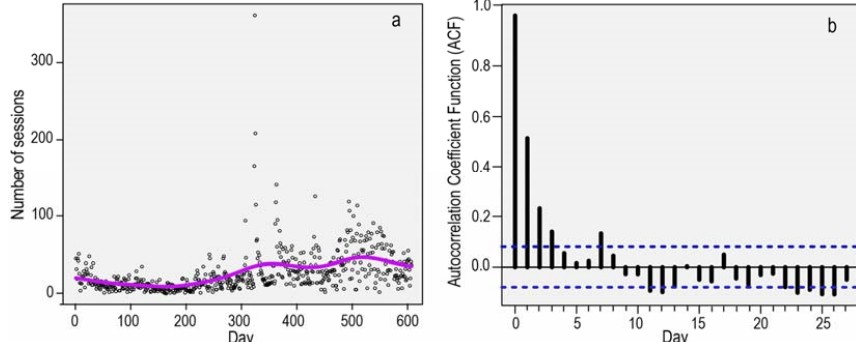




Natural Hazards
and Earth System
Figure 6. Daily statistics of the number of users (a) and a comparison between the pageviews and the number
of users (b) for the 21 months of site activity in the 16 January 2014 – 15 October 2015 period.

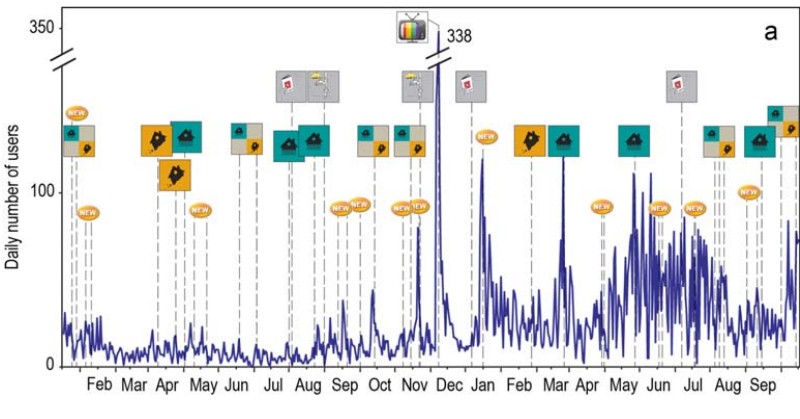

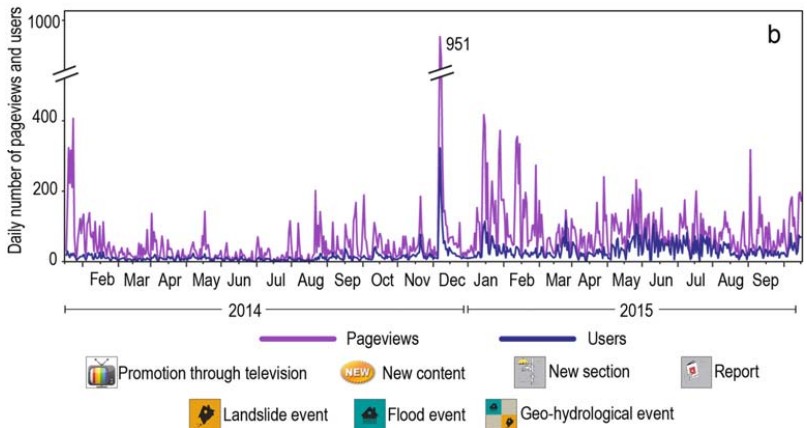






Figure 7. The plot shows the number of unique Facebook page users. In the plot, the days when the extreme
rainfall condition occurred were reported with blue dots; the major geo-hydrological event occurrences were
reported in orange, and the severe warning declarations were reported with red dots.

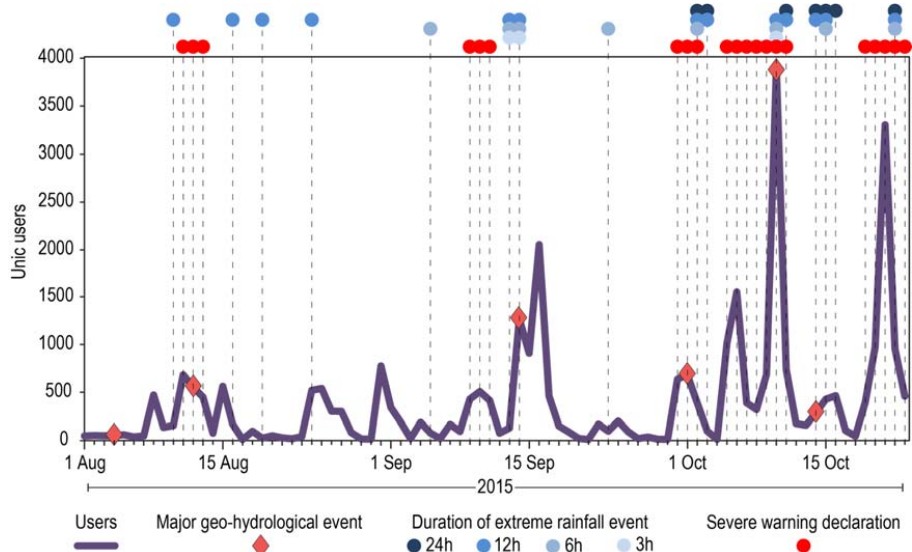
