# Peer review of "Communication strategies to address geo-hydrological risks"

_Natural Hazards and Earth System Sciences, 2015_

## Referee Comment (RC1) · M. Bostenaru-Dan (Referee) · 6 Feb 2016

The paper presents a valuable approach of citizent participation into disaster management. The paper properly recognises the contemporary dimension of participation, which is communication of the appropriate event and connected intervention. However, the relationship between participation and communication is not properly explained with the sociological background. For a solid view on the sociology of disasters I would recommend the studies by Plapp (or Kunz-Plapp), some of them in NHESS. Some of them refer to flooding.

A proper communication is influencing the perception of risk, and this is also a link I miss in the paper. Also, some studies on risk perception from a psychology background are done by the team around Armas, from which those by Nenciu (Posner) on flood,

available in Natural Hazards recently. Also Krebich published in NHESS and elsewhere on perception of flood and how it leads to measures in Germany. Myself I presented the sociology background of participation and communication for earthquakes in NHESS.

From the scientific content I would be interested how the citizens interact, if they are just foreseen with information (first step of participation, see my article) or if they can provide information, ex. through mobile devices, as designed in several EU projects for continuing the information gathering after the project end. Recently ESRI started a MOOC of teaching creating story maps which do not require programming knowledge. It seems to me that only the blog section is bottom up. However, what differentiates communication and participation is how citizens can engage more than perceiving. Can they contact the civil protection to organise to bring sand bags for example? In Bernburg in 2013 I assisted to successful prevention of flood damage despite flood due to citizen engagement. (I attach the photo of protection, which was effective, in Halle on the same river Saale the dam broke) Over the information of citizen, citizens must become active to do constructive participation, not only protest in case what unliked events and perception. Is this development foreseen in the future for the product if not already existing?

I hope this helps to improve the paper.

––––––––––––––––––––––––

[Figure]

[Figure]

**Fig. 1.** flood prevention through citizen intervention in Bernburg 2013

---

## Referee Comment (RC2) · G. Bambara (Referee) · 26 Feb 2016

**GENERAL COMMENTS**

The paper is well written and very understandable. It presents an interesting tool (PO-LARIS) to inform the mass media and citizens on landslides and floods. For this reason, this paper is appropriate for the journal NHESS. Explanation of the website information architecture is well done. The analysis that is done and the results are interesting and show the interest of this study. Figures are of good quality and well explained. For me, this paper is a first analysis of POLARIS site. We logically expect a second paper which will give more results, for example with concrete feedback about the influence of this type of site on the resilience society during a hazard event. Thus, in this paper it would have been well to develop a little more discussion and what is

planned to do next. Moreover, it lacks a discussion on comparison with what is done and achieved in other countries. Finally, it is important to give the prospects of this study: a Computer-Assisted Telephone Interviews was conducted in 2012 and 2013, is that an investigation is planned? POLARIS website, has been helpful? It would be nice to do this analysis at least in areas where has increased site traffic. This would verify the effectiveness of the site for information to the public

SPECIFIC COMMENTS

Introduction

l. 74 at 82 - when is it in other countries? l. 76 - remove space

POLARIS website

l.108 - As there is no section 3.2, section 3.1 seems to me unnecessary and could be grouped in section 3. The title of the latter (section 3) can be refined.

---

## Author Comment (AC1) · 27 Apr 2016

REV#1 The paper presents a valuable approach of citizen participation into disaster management. The paper properly recognises the contemporary dimension of participation, which is communication of the appropriate event and connected intervention.

We thank the reviewer for this comment.

However, the relationship between participation and communication is not properly explained with the sociological background.

We accepted this interesting suggestion, and added a new chapter: 2. Background in risk communication and perception, for analysing the state of art about the public participation in the context of natural hazard risk communication. The new text reads

(lines 89-109): "Extensive discussions have been occurred in the past about the most appropriate ways to manage the potential consequences of natural hazards (Scolobig et al. 2015), and governments began to institutionalize disaster risk management processes and practices (McEntire, 2006). More recently, an integrated approach to risk management processes is emerging, encompassing in a coordinated way activities needed to preserve a level of safety with regard the risk posed by natural hazards (http://www.climchalp.org/). Initially associated with environmental management, public health, and emergency management matter, risk communication aims at informing people about a potential hazard and the associated harms (Steelman and McCaffrey, 2012). In the last decade, the relevance of communication is increasing in response to the changes affecting risk governance (Höppner et al., 2010). Accordingly, communication must serve multiple purposes spanning all phases of risk management (Renn 2005) enabling more effective decisions, knowledge-based actions (Höppner et al., 2010), and addressing the exchange of knowledge and attitudes between all the involved actors (i.e., public bodies, private sectors, third sector, citizens). In this context, public participation is crucial, and defined as the co-decision in planning processes designed by others, where the central elements of the participation concept are influence, interaction, and information exchange (Bostenaru, 2004). Starting in the 1990s, extensive public consultation and participation in risk management have focused on re-establishing public trust (Rowe et al., 2004). The appropriate transfer of knowledge between experts and the broader public can be facilitated by effective communication strategies and programs, at national or local level, to align the views of the public with those of the experts (Frewer, 2004). More recently, the increased attention of public institutions to stimulate the participation of citizens in the definition and delivery of public services is leading to the adoption of a citizen-centred risk management approach which takes into account social concerns and the citizens' s perception about risks." For a solid view on the sociology of disasters I would recommend the studies by Plapp (or Kunz-Plapp), some of them in NHESS. Some of them refer to flooding.

We thank for the suggestion and we integrated the findings of the recommended studies in the new chapter of background.

A proper communication is influencing the perception of risk, and this is also a link I miss in the paper. Also, some studies on risk perception from a psychology background are done by the team around Armas, from which those by Nenciu (Posner) on flood available in Natural Hazards recently. Also Krebich published in NHESS and elsewhere on perception of flood and how it leads to measures in Germany. Myself I presented the sociology background of participation and communication for earthquakes in NHESS. We accepted this suggestion, and we integrated the added chapter with a brief background in risk perception of natural hazards. The new text reads (lines 110-134): "Risk perception is also important to determine the attitude towards risks and, when information campaigns and risk communication strategies are designed, the public perception should be known (Plapp & Werner, 2006). Risk perception is a subjective assessment of the hazard occurrence's probability and people's feelings of the consequences (Posner & Armas 2014). A gap between the public's perception of their own responsibility, and that of authorities in terms of risk reduction was found by Fernández-Bilbao and Twigger-Ross (2009) who, working in England and Germany, found that the public did not perceive that reducing flood risk was their responsibility. Plattner et al. (2006) highlighted a systematic discrepancy between the individual subjective risk evaluation (perceived), and formal risk evaluation procedures. Similarly, in Italy two national surveys conducted to measure the public perception of landslide and flood risk confirmed that in most of the Italian regions the observed perception of the threat did not match the long-term risk posed by landslides and floods to the population (Salvati et al., 2014). If it is globally accepted that risk perception has strong implication for the success of risk communication. It is also expected that effective risk communication shapes risk perception (Höppner et al., 2010). There are many studies trying to establish which formats of communication may be most effective (e.g., Faulkner and Ball 2007; Fernandez- Bilbao and Twigger-Ross 2009; Kashefi and Walker 2009; Bier 2001). Three phases of risk communication were identified by Leiss (1996) in the USA, including one-way communication, persuasive communication, and two-way

communication. As Höppner et al. (2010) reported, the first is primarily used to convey probabilistic information, educate the public at risk, and to gain consent over risk management practices, whereas the second is thought to change people's risk related behaviours. In the latter phase, all actors should engage with, and learn from each other (Renn, 2005). Risk communication is a complex activity moving from the one-way distribution of information towards a two-way exchange of knowledge and more participatory approach (Höppner et al., 2010). Despite this latter communication approach seems to be more effective, in the review work conducted by Höppner et al. (2010) between all the communication practices posed by governmental authorities, national and local agencies, the majority resulted one-way efforts, focused solely on improving hazard knowledge or raising risk awareness, mostly regarding flood hazard."

From the scientific content I would be interested how the citizens interact, if they are just foreseen with information (first step of participation, see my article) or if they can provide information, ex. through mobile devices, as designed in several EU projects for continuing the information gathering after the project end. Recently ESRI started a MOOC of teaching creating story maps which do not require programming knowledge.

The Polaris initiative does not have "teaching" among its goals; and as such, it does not consider any e-learning (e-teaching) activity. Devoted to increase awareness on risk perception, we reckon that the provision of information (i.e. images, comments) from citizens through Polaris should be stimulated by improving the integration of the website content with social media such as Istagram and Pinterest where people prefer to exchange images and content instead of using websites.

It seems to me that only the blog section is bottom up. However, what differentiates communication and participation is how citizens can engage more than perceiving. We agreed with the reviser comment and to better explain the role the blog section and the relation between Polaris and the public we added a new paragraph for discussion. The text now reads (lines 353-381): "In Polaris we mean risk communication as a two-way exchange of related information and knowledge on natural hazards and associated risk

for the population. The Blog section of the website is mainly encouraging bottom-up feedback through visitors' s comments. The link to Facebook stimulates more feedback from citizens who upload pictures and make post on Facebook. This means that participation, whose central elements are influence, interaction and information exchange (Bostenaru, 2004), is mainly facilitated by the link with Facebook. However, the website Blog section remains less active than we expected, for at least two reasons: first, in Italy, the perception of geo-hydrological hazards is still very weak, people show less interested toward these geo-hydrological events than to other natural hazards such as, seismic risk (Salvati et al., 2014). Second, people do not know how a geo-hydrological event can hit them. People are interested to actively participate through the blog section mainly when a particularly disastrous event is occurring, and in such a case, by simply uploading videos and pictures rather than asking for explanation or advices. This means that, despite many institutions are making efforts to increase the public understanding of geo-hydrological risk through nationwide awareness campaigns (e.g. I do not risk, http://iononrischio.protezionecivile.it/), people still ignore how a large part of the Italian territory suffers of geo-hydrological risk. Such an underestimation of the possible risks, the high confidence in the local administrators towards which citizens delegate their personal safeness are all factors that impede an effective risk communication. It is important to highlights that Polaris offers a knowledge-oriented risk communication which tends to operate continuously and does not regard the warning messages released in the event of a disaster. The communication efforts seeks to change the people's attitudes to the geo-hydrological hazard that they may have encountered giving many examples of what had happened before. People will not react to risk warnings if foregoing communication has not motivated and prepared them. For this purpose, we are going to evolve the Blog section of Polaris which is the most relevant for stimulating public participation at any moment. In particular, we plan to integrate other relevant social media, such as Instagram and Pinterest, stimulating the sharing of images and videos and the associated tags and comments. For encourage more resilient behaviours during the occurrences of hazardous events, we would stimulate

the usage of video through the YouTube and Vimeo channels that we can comment for feedback and/or advices. Finally, we are going to create new synergies with the "I do not risk" campaign and website of the Italian Department of Civil Protection, which will increase traffic, information exchange and, as such, strengthen the risk perception by the Italian population."

Can they contact the civil protection to organise to bring sand bags for example? In Bernburg in 2013 I assisted to successful prevention of flood damage despite flood due to citizen engagement. (I attach the photo of protection, which was effective, in Halle on the same river Saale the dam broke) Over the information of citizen, citizens must become active to do constructive participation, not only protest in case what unlike events and perception. Is this development foreseen in the future for the product if not already existing?

We agree with the reviewer opinion but this is not in the scope of Polaris. The website project is an initiative developed by a research institution and not by the public institutions in charge of risk management (e.g. national and/or local civil protection authorities).

I hope this helps to improve the paper. We thank the reviewer for the helpful in improving the paper.

Please also note the supplement to this comment:
http://www.nat-hazards-earth-syst-sci-discuss.net/nhess-2015-354/nhess-2015-354-AC1-supplement.pdf

---

## Author Comment (AC2) · 27 Apr 2016

The paper is well written and very understandable. It presents an interesting tool (PO-LARIS) to inform the mass media and citizens on landslides and floods. For this reason, this paper is appropriate for the journal NHESS. Explanation of the website information architecture is well done. The analysis that is done and the results are interesting and show the interest of this study. Figures are of good quality and well explained. For me, this paper is a first analysis of POLARIS site.

We thank the reviewer for this comment.

We logically expect a second paper which will give more results, for example with concrete feedback about the influence of this type of site on the resilience society during a hazard event.

We accepted this interesting and challenging suggestion for the future of Polaris. After two years from the first analysis of the risk perception of geo-hydrological risk by the Italian population carried out in 2012 and 2013, we are testing once again the risk perception through an on line survey (https://goo.gl/6cetpm) addressed at the schools (students, administrative and technical staff, teachers, parents, etc.) in the Umbria region in Italy. This will help to assess the possible change in the public perception concerning geo-hydrological risk. We plan to publish the results in a new paper.

Thus, in this paper it would have been well to develop a little more discussion and what is planned to do next.

Suggestion accepted, we integrated the manuscript with a new chapter where some discussion are reported and where we added our plans for the future. The text reads (lines 374-381): "For this purpose, we are going to evolve the Blog section of Polaris which is the most relevant for stimulating public participation at any moment. In particular, we plan to integrate other relevant social media, such as Instagram and Pinterest, stimulating the sharing of images and videos and the associated tags and comments. For encourage more resilient behaviours during the occurrences of hazardous events, we would stimulate the usage of video through the YouTube and Vimeo channels that we can comment for feedback and/or advices. Finally, we are going to create new synergies with the "I do not risk" campaign and website of the Italian Department of Civil Protection, which will increase traffic, information exchange and, as such, strengthen the risk perception by the Italian population". Moreover, it lacks a discussion on comparison with what is done and achieved in other countries.

We accepted the suggestion. We integrated the manuscript adding a new chapter titled "2. Background in risk communication and perception", where we also reported briefly what was done in other countries concerning communication of natural hazards. The text reads (lines 89-134): "Extensive discussions have been occurred in the past about the most appropriate ways to manage the potential consequences of natural hazards (Scolobig et al. 2015), and governments began to institutionalize disaster risk

management processes and practices (McEntire, 2006). More recently, an integrated approach to risk management processes is emerging, encompassing in a coordinated way activities needed to preserve a level of safety with regard the risk posed by natural hazards (http://www.climchalp.org/). Initially associated with environmental management, public health, and emergency management matter, risk communication aims at informing people about a potential hazard and the associated harms (Steelman and McCaffrey, 2012). In the last decade, the relevance of communication is increasing in response to the changes affecting risk governance (Höppner et al., 2010). Accordingly, communication must serve multiple purposes spanning all phases of risk management (Renn 2005) enabling more effective decisions, knowledge-based actions (Höppner et al., 2010), and addressing the exchange of knowledge and attitudes between all the involved actors (i.e., public bodies, private sectors, third sector, citizens). In this context, public participation is crucial, and defined as the co-decision in planning processes designed by others, where the central elements of the participation concept are influence, interaction, and information exchange (Bostenaru, 2004). Starting in the 1990s, extensive public consultation and participation in risk management have focused on re-establishing public trust (Rowe et al., 2004). The appropriate transfer of knowledge between experts and the broader public can be facilitated by effective communication strategies and programs, at national or local level, to align the views of the public with those of the experts (Frewer, 2004). More recently, the increased attention of public institutions to stimulate the participation of citizens in the definition and delivery of public services is leading to the adoption of a citizen-centred risk management approach which takes into account social concerns and the citizens' s perception about risks. Risk perception is also important to determine the attitude towards risks and, when information campaigns and risk communication strategies are designed, the public perception should be known (Plapp & Werner, 2006). Risk perception is a subjective assessment of the hazard occurrence's probability and people's feelings of the consequences (Posner & Armas 2014). A gap between the public's perception of their own responsibility, and that of authorities in terms of risk reduction was found by

Fernández-Bilbao and Twigger-Ross (2009) who, working in England and Germany, found that the public did not perceive that reducing flood risk was their responsibility. Plattner et al. (2006) highlighted a systematic discrepancy between the individual subjective risk evaluation (perceived), and formal risk evaluation procedures. Similarly, in Italy two national surveys conducted to measure the public perception of landslide and flood risk confirmed that in most of the Italian regions the observed perception of the threat did not match the long-term risk posed by landslides and floods to the population (Salvati et al., 2014). If it is globally accepted that risk perception has strong implication for the success of risk communication. It is also expected that effective risk communication shapes risk perception (Höppner et al., 2010). There are many studies trying to establish which formats of communication may be most effective (e.g., Faulkner and Ball 2007; Fernandez- Bilbao and Twigger-Ross 2009; Kashefi and Walker 2009; Bier 2001). Three phases of risk communication were identified by Leiss (1996) in the USA, including one-way communication, persuasive communication, and two-way communication. As Höppner et al. (2010) reported, the first is primarily used to convey probabilistic information, educate the public at risk, and to gain consent over risk management practices, whereas the second is thought to change people's risk related behaviours. In the latter phase, all actors should engage with, and learn from each other (Renn, 2005). Risk communication is a complex activity moving from the one-way distribution of information towards a two-way exchange of knowledge and more participatory approach (Höppner et al., 2010). Despite this latter communication approach seems to be more effective, in the review work conducted by Höppner et al. (2010) between all the communication practices posed by governmental authorities, national and local agencies, the majority resulted one-way efforts, focused solely on improving hazard knowledge or raising risk awareness, mostly regarding flood hazard".

Finally, it is important to give the prospects of this study: a Computer-Assisted Telephone Interviews was conducted in 2012 and 2013, is that an investigation is planned? POLARIS website, has been helpful? It would be nice to do this analysis at least in areas where has increased site traffic. This would verify the effectiveness of the site

for information to the public

POLARIS website was published in 2014, after the surveys that we conducted in 2012 and 2013 using a Computer-Assisted Telephone Interviews. The two surveys were executed in collaboration with DOXA, a leading Italian company operating in the field of statistical research and opinion polls. The survey results were published in a paper (Salvati et al, 2014 Perception of Flood and Landslide Risk in Italy: a Preliminary Analysis. Nat. Hazards Earth Syst. Sci., 14, 2589-2603,) and in a dedicated focus on Polaris (http://polaris.irpi.cnr.it/la-percezione-del-rischio-da-frana-e-inondazione-in-italia/). Findings from the analysis of the perception of the Italian population motivated our purpose to increase the awareness of the Italian population and influenced the structure of the content of Polaris. We are now testing once again the risk perception through an on line survey (https://goo.gl/6cetpm) addressed at the schools (students, administrative and technical staff, teachers, parents, etc.) in the Umbria region in Italy. Our idea is to propose the same survey to other Italian regions. This will help to assess the possible change in the public perception concerning geo-hydrological risk.

Please also note the supplement to this comment:
http://www.nat-hazards-earth-syst-sci-discuss.net/nhess-2015-354/nhess-2015-354-AC2-supplement.pdf

---

## Author Comment (AC3) · 27 Apr 2016

Dear Editor, This cover letter accompanies the answers to referee comments for the manuscript: Communication strategies to address geo-hydrological risks: the POLARIS web initiative in Italy by Paola Salvati, Umberto Pernice, Cinzia Bianchi, Federica Fiorucci, Ivan Marchesini and Fausto Guzzetti. We are very much grateful to you and to the two reviewers for the constructive comments and suggestions that helped us to improve the work.

In particular we would like to summarize that:

1) to explain the relationship between participation and communication and the link between communication and perception we: (i) modified the Introduction, (ii) added a new paragraph entitled "Background in risk communication and perception" and (iii) added

a new paragraph "Discussion"; 2) in the background we analysed the literature in risk communication and risk perception in the context of natural hazards and in the discussion we considered the website Polaris in the light of the current meaning of risk integrated management approach; 3) we provided a list of our responses to the referee's comments, including the track changes made to the text; 4) we changed the term "Are you Ready" into "Are you Prepared" (the English translation of "Sei Preparato" section). Consequently, we modified and uploaded the new fig. 2 showing the POLARIS Home Page; 5) we made some minor improvements of the English language.

Overall, we consider this new version of the manuscript significantly improved and we enclose the new pdf file of the manuscript as supplement file.

We thank you and wait for further communication from you.

Sincerely, Paola Salvati, on behalf of all authors

Please also note the supplement to this comment:
http://www.nat-hazards-earth-syst-sci-discuss.net/nhess-2015-354/nhess-2015-354-AC3-supplement.pdf

**Supplement:**

**Communication strategies to address geo-hydrological risks:**

**the POLARIS web initiative in Italy**

P. Salvati[1], U. Pernice[2], C. Bianchi[1], I. Marchesini[1], F. Fiorucci[1], F. Guzzetti[1]

(1) Consiglio Nazionale delle Ricerche, Istituto di Ricerca per la Protezione Idrogeologica, via Madonna Alta 126, I-06128
Perugia, Italy (2) Innovation Consultant, Viale Michelangelo 2315, I-90135 Palermo, Italy (*) *Corresponding author*, Paola.Salvati@irpi.cnr.it, Tel. +39 075 50144427, Fax +39 075 5104420

**Abstract.** Floods and landslides are common phenomena that cause serious damage and pose a severe threat to the population of Italy. The societal and economic impact of floods and landslides in Italy is severe, and strategies to target the mitigation of the effects of these phenomena are needed. In the last few years, the scientific community has started to use web technology to communicate information on geo-hydrological hazards and the associated risks. However, the communication is often targeted to technical experts. In the attempt to communicate to a broader audience relevant information on geo-hydrological hazards with potential human consequences to the population, we designed the POLARIS website. POLARIS publishes accurate information on geo-hydrological risk to the population of Italy, including periodic reports on landslide and flood risk to the population, analyses of specific damaging events, and blog posts on landslide and flood events. By monitoring the access to POLARIS in the 21-month period between January 2014 and October 2015, we found that access increased during particularly damaging geo-hydrological events and immediately after the web site was advertised by press releases. POLARIS demonstrates that the scientific community can implement suitable communication strategies that address different societal audiences, exploiting the role of mass media and social media. The strategies can help multiple audiences understand how risks can be reduced through appropriate measures and behaviors, contributing to increasing the resilience of the population to geo-hydrological risk.

**1  Introduction**

Geo-hydrological hazards, including floods and landslides, are common geo-hydrological phenomena that cause serious damage and pose severe threats to the population worldwide. Currently, river flooding annually affects 21 million people worldwide, and the estimate is expected to reach 54 million people by 2030 (http://www.wri.org). For landslides, Petley (2012) showed that human losses were considerably higher than had been previously considered. Global costs of geo-hydrological disasters have increased in recent decades and, in future decades, it is expected that the number of people at risk and the occurrence of extreme events will both grow (https://www.ipcc.ch). Integrated risk management involving public authorities, research scientists, companies, and citizens is required to address the interconnectivity between physical infrastructures, economic systems and the role of human factors (Jonkman and Dawson, 2012). The approach should encompass, in a coordinated way, all the necessary activities to maintain a level of security with regard the risk posed by natural hazards (http://www.climchalp.org/) including exchange of information and experience between public bodies, business bodies and citizens.

The availability of detailed and organized information on the geographical and temporal distribution of geo-hydrological events and their consequences, communicated throughout different media channels, is important to implement national communication strategies and preparedness programs. In Italy, detailed information on landslides and floods is available, and catalogues of landslide and flood events with fatalities have been organized and constantly updated (Guzzetti et al., 1994, 2005; Guzzetti and Tonelli, 2004; Salvati et al., 2010, 2012, 2013). For this country, in recent decades, much effort has been exerted to analyse landslide and flood hazards and the associated risk at various geographical scales, from the site specific (local) to the synoptic (national) scale. Despite these efforts, most of these studies remain unknown to the public, that ignores the possible damaging effects that landslides and floods can produce (Salvati et al., 2014). Despite the large number and wide geographical distribution of landslide and flood events, the Italian population receives minimal information and has minimal knowledge on the type, characteristics, frequency, and severity of the harmful events that have occurred in the area where they live, or work. The lack of knowledge is amplified by a weak motivation of the people to be informed and as a consequence they demonstrate weak understanding and perception of geo-hydrological risk (Salvati et al., 2014).

Although, in the last few years, the Italian scientific community has begun to communicate information on geo-hydrological hazards and the associated risks through communication initiatives and thematic websites (http://avi.gndci.cnr.it/; http://sici.irpi.cnr.it/; http://www.isprambiente.gov.it/it/progetti/suolo-e-territorio-1/iffi-inventario-dei-fenomeni-franosi-in-italia; http://www.pcn.minambiente.it/GN/), these often suffer from the lack of effective communication strategies capable of addressing various targets with suitable media. Consequently, the initiatives remain addressed mainly to experts, for specific technical purposes with content and web interfaces that are barely appreciated by a wider audience, and rarely synchronized with social media networks.

Various problems emerged when designing the communication strategy. First, public interest in the issue is important. As Keys (1999) noted, "*It has been apparent for some time that creating community awareness of*

*floods and storms is not easy, (....) Most of the time, people are not particularly interested in them*" (O'Neil,

2004). The core of the problem is to capture public attention and, with long-term actions, familiarise people to the topic. Knowledge-oriented risk communication campaigns on the causes and dynamics of geo-hydrological hazards and their possible consequences to human life, conducted with appropriate frequency, can effectively increase public awareness of geo-hydrological hazards. Second, it is important to find the appropriate mediators to reach the largest number of people. Media represent key mediators of communication between different audiences i.e., the public, scientists, policy-makers, and the operational management (Beck, 1992). They act as

*social glue* with respect to the perception and interpretation of natural hazards in heterogeneous societies (Miles and Morse, 2007).

The mission of the POLARIS website is to provide correct and reliable information mainly to media, which will help to further communicate the information to other audiences. In addition, the role of social media should be carefully considered to engage audiences that are typically weakly interested in information on geo- hydrological risk. Thus, efforts were made to improve the link between the POLARIS website and the Facebook page (https://www.facebook.com/CNR.IRPI) of the Istituto di Ricerca per la Protezione Idrogeologica (IRPI, http://www.irpi.cnr.it), of the Italian Consiglio Nazionale delle Ricerche (CNR, http://www.cnr.it), by conveying immediate and concise information on natural disasters using pictures and videos, interspersed with invitations to visit the POLARIS website for detailed information.

Following an overview of the literature on natural hazard's risk communication, in this paper, we describe the website information architecture; we analyse the users' navigation data during the 21-month period since the website was published. Then, we explain possible relations between the maximum access and the context in which they occurred. Finally, we discuss possible future improvement of the site and conclude by summarizing our findings.

**2    Background in risk communication and perception**

Extensive discussions have been occurred in the past about the most appropriate ways to manage the potential consequences of natural hazards (Scolobig et al. 2015), and governments began to institutionalize disaster risk management processes and practices (McEntire, 2006). More recently, an integrated approach to risk management processes is emerging, encompassing in a coordinated way activities needed to preserve a level of safety with regard the risk posed by natural hazards (http://www.climchalp.org/). Initially associated with environmental management, public health, and emergency management matter, risk communication aims at informing people about a potential hazard and the associated harms (Steelman and McCaffrey, 2012). In the last decade, the relevance of communication is increasing in response to the changes affecting risk governance (Höppner et al., 2010). Accordingly, communication must serve multiple purposes spanning all phases of risk management (Renn 2005) enabling more effective decisions, knowledge-based actions (Höppner et al., 2010), and addressing the exchange of knowledge and attitudes between all the involved actors (i.e., public bodies, private sectors, third sector, citizens). In this context, public participation is crucial, and defined as the co- decision in planning processes designed by others, where the central elements of the participation concept are influence, interaction, and information exchange (Bostenaru, 2004). Starting in the 1990s, extensive public consultation and participation in risk management have focused on re-establishing public trust (Rowe et al.,

2004). The appropriate transfer of knowledge between experts and the broader public can be facilitated by effective communication strategies and programs, at national or local level, to align the views of the public with those of the experts (Frewer, 2004). More recently, the increased attention of public institutions to stimulate the participation of citizens in the definition and delivery of public services is leading to the adoption of a citizen- centred risk management approach which takes into account social concerns and the citizens' s perception about risks.

Risk perception is also important to determine the attitude towards risks and, when information campaigns and risk communication strategies are designed, the public perception should be known (Plapp & Werner, 2006).

Risk perception is a subjective assessment of the hazard occurrence's probability and people's feelings of the consequences (Posner & Armas 2014). A gap between the public's perception of their own responsibility, and that of authorities in terms of risk reduction was found by Fernández-Bilbao and Twigger-Ross (2009) who, working in England and Germany, found that the public did not perceive that reducing flood risk was their responsibility. Plattner et al. (2006) highlighted a systematic discrepancy between the individual subjective risk evaluation (perceived), and formal risk evaluation procedures. Similarly, in Italy two national surveys conducted to measure the public perception of landslide and flood risk confirmed that in most of the Italian regions the observed perception of the threat did not match the long-term risk posed by landslides and floods to the population (Salvati et al., 2014).

If it is globally accepted that risk perception has strong implication for the success of risk communication. It is also expected that effective risk communication shapes risk perception (Höppner et al., 2010). There are many studies trying to establish which formats of communication may be most effective (e.g., Faulkner and Ball

2007; Fernandez- Bilbao and Twigger-Ross 2009; Kashefi and Walker 2009; Bier 2001). Three phases of risk communication were identified by Leiss (1996) in the USA, including one-way communication, persuasive communication, and two-way communication. As Höppner et al. (2010) reported, the first is primarily used to convey probabilistic information, educate the public at risk, and to gain consent over risk management practices, whereas the second is thought to change people's risk related behaviours. In the latter phase, all actors should engage with, and learn from each other (Renn, 2005). Risk communication is a complex activity moving from the one-way distribution of information towards a two-way exchange of knowledge and more participatory approach (Höppner et al., 2010). Despite this latter communication approach seems to be more effective, in the review work conducted by Höppner et al. (2010) between all the communication practices posed by governmental authorities, national and local agencies, the majority resulted one-way efforts, focused solely on
improving hazard knowledge or raising risk awareness, mostly regarding flood hazard.

**3    Nomenclature**

In this work, we adopt the terminology and definitions used in Google Analytics. We use the term *session* to
indicate the period of time a user is actively engaged with the POLARIS website. All usage data (screen views,
events, ecommerce) are associated to a session. *Users* are people who have had at least one session in the
selected date range, including new and returning users. *Pageviews* are the total number of pages viewed,
including repeated views of the same page. The *source* is the place users were before viewing a POLARIS
website content, including a search engine or another website. *Referral traffic* is Google's method of reporting
visitors that arrived at a specific site from sources outside their search engine.

**4    POLARIS website**

The effectiveness of the POLARIS communication strategy relies on the main assumption that the scientific
community can play a key role in increasing awareness (Bier, 2001) of individuals and groups on geo-
hydrological hazards, and on the type and extent of the risk posed by geo-hydrological hazards to the population.
This role should be attained working in two directions: (i) providing mass media (e.g., journalists) with correct
and reliable information, which they can communicate (spread) further to the broader civil society, and (ii)
adopting less technical and more widely comprehensible language to better engage citizens. Figure 1 shows the
communication flow adopted in POLARIS, where the scientists use different communication approaches to
mass media, civil protection and local/regional authorities, and to citizens. In this framework, the media captures
information from scientists and uses it for communication purposes.

The scientific and technical content of POLARIS is based on a communication strategy that avoids scientific
and technical terminology, in favour of a more widely understandable language. For this purpose, consultants
experienced in web-communication strategies on natural hazards, info-graphics, and user experience design
were involved in the initiative. The consultants' contribution consisted in arranging the messages using intuitive
and engaging web interfaces to display data, graphs, tables, video and in carefully considering usability and
accessibility of the website to diversified audiences.

POLARIS is based on a well-defined information architecture encompassing six main sections: (i) Reports, (ii)
Are you prepared?, (iii) Events, (iv) Alert Zones, (v) Focus, and (vi) Blog. The sections provide different and
complementary information, including: (i) periodical reports with analyses of landslide and flood risk to the
population of Italy, (ii) suggestions on suitable behaviours to adopt before, during and after potentially
damaging events, (iii) data and synthetic analyses of specific geo-hydrological events with human
consequences, (iv) visual information on the morphology, geology, and historical damaging events of the Alert

Zones used by the Italian Civil Protection system for issuing warning on meteorological, hydrological, and
geomorphological hazards, (v) detailed analyses of relevant topics or specific events with severe consequences,
and (vi) blog-posts on landslide and flood events aimed at encouraging citizens' engagement. Fig. 2 shows the
POLARIS home page, with specifically-designed images and graphics to help browse the website.

## 4.1    Structure of the POLARIS website

The "Reports" section illustrates periodic reports on landslide and flood risk to the population of Italy. Reports
are published every six months. The last report is available in two formats: (i) an on-line version, and (i) a
standard Adobe® PDF (Portable Document Format) file. The on-line report is directly integrated with the CNR
IRPI Spatial Data Infrastructure, SDI (Salvati et al., 2013) where the database is located, and has access to data
kept updated regularly. Each report contains the list of landslides and floods that occurred in the period (six
months, or a year), with information on the date, location, dead and missing persons, injured people, maps,
statistics, and an analysis of the landslide and flood events with direct consequences to the population. Statistics
are available for different periods of one, five, and fifty years, enabling comparative analyses of the
geographical and temporal variations of geo-hydrological risk in Italy.

The "Events" section publishes information on specific meteorological events in Italy, using text, maps, videos,
photographs, and drawings. In this section, specific icons were designed to define the type of the geo-
hydrological events. A short text containing information on the sites affected, the damage, and the fatalities or
casualties is given, with a map showing the location of landslide and flood that affected the population. The
"Focus" section publishes information on specific topics, provides analysis for each Italian region, and offers
descriptions of single historical or recent catastrophic geo-hydrological events. The "Events" and "Focus"
sections jointly inform the population on the extent and severity of geo-hydrological risk in Italy. They also
represent an important source of information and data for the mass media.

The "Alert zones" section provides information for 134 Alert Zones defined by the Italian National Civil
Protection system to forecast geo-hydrological hazards, including landslides and floods. The section provides
the possibility to query a number of information items, and a sidebar offers access to different thematic layers
and maps for each Alert Zone.

The "Are you prepared?" section offers information on suitable (and unsuitable) behaviours to adopt before,
during, and after a damaging geo-hydrological event. The suggested elementary behavioural rules may save
people's lives.

Finally, the "Blog" section encourages bottom-up participation by users, who can post comments on geo-
hydrologic hazards and risks.

In the home page, particular focus is reserved to a section called "It Happened Today" (Italian: *Accadde oggi*),
which is a daily register of events in which, for each day of the year, POLARIS publishes a short description of relevant events that adversely impacted the population that specific day. This section is directly linked to the
CNR IRPI SDI, which daily automatically relates the event to the exact day.

**5    Data**

We use Google Analytics to monitor the traffic and performance of the POLARIS website, focusing our analysis
on (i) channels used, (ii) number of sessions, (iii) number of users, (iv) users viewing single pages or the entire
website, and (v) the geographical distribution of the users. We further monitored POLARIS' Facebook page
using "Insight" instrument and particularly the number of "likes" given by users, or the number of users who
viewed the posts. We also performed an analysis of the type of posts (containing video, link, images, or text
alone) that interested more the users, and their origin.

**6    Analysis and results**

In this section, we describe the analysis performed to identify possible trends of interest to the POLARIS
content, and the dependence between peak access values to the website and possible causes that increased the
public interest in the website. We also performed similar analysis for the CNR IRPI Facebook page, which is
the Institute's most active social network.

**6.1    POLARIS website**

The analysis of the data series available from Google Analytics for the period of the website publication, from
16 January 2014 to 15 October 2015, allowed to prepare general statistics summarized in **Table 1**, where we
listed the data separately for sessions, users, pageviews, and referrals from social networks. We studied the
geographical distribution of the users, and the number of pageviews for each section of the website. Results are
shown in Fig 3.

Since POLARIS is published in Italian, it is not surprising that the sessions mainly originate from Italy (91%).
Figure 3a shows the geographical distribution of the sessions in Italy. The limited percentage of sessions
originating from other nations concentrates in the USA, China, Japan, and Germany. Darker and larger dots in
the map show the increasing number of sessions, with few areas where sessions are highly concentrated. The
largest number of sessions originate from Umbria, where the main office of CNR IRPI is located. Other areas
from where POLARIS was accessed frequently include Rome, where the majority of the government offices
are located, Milan (Lombardy), Turin (Piedmont), Genoa (Liguria) and Palermo (Sicily). These cities host
institutes and researchers who are interested in geo-hydrological issues. Collectively, they also host 6 million
people, 10% of the entire population of Italy.

The pie chart in Fig. 3 shows the number of pageviews for the different sections of the website. Not surprisingly,
the home page is the most viewed page, containing, in addition to the navigation menu, the "It Happened Today"

(*Accadde Oggi*) section, which is read by many people, most probably because the content changes daily. The
second most viewed section is the Report section, which publishes periodic reports on the risks posed to the
Italian population by landslides and floods. This section is updated every six months, and allows to download
the reports as PDF files. The "Focus" and "Event" sections have similar access percentages. Their content is
simple to read and straightforward to understand thanks to explicative figures and maps. The content differs in
the subjects; on the Focus page, we discuss in-depth issues related to geo-hydrological hazards and risks,
whereas the Events section is dedicated to the description of specific events that caused damages to the Italian
population. The "Alerts Zones" and "Are You Prepared?" sections were not accessed as much as expected,
although they both contain relevant information and suggestion to help develop suitable behaviours toward
disaster resilience.

Monitoring the number of sessions during the 21 months since the website's publication, it was possible to
study their temporal distribution. For the purpose, we normalized the number of sessions per day to the daily
average number of sessions in the 21-month period (long-term average, 26.9). Results are shown in Fig. 4,
where the ratio in the x-axis represents the daily access number divided by the average access number in the
observation period. The grey parts of the line show periods below the long-term average, and the blue parts
show periods above the long-term average. Inspection of Fig. 4 reveals that there was an increase in the number
of sessions (blue dashed line in Fig. 4) and significant variations in the daily distribution are also evident. We
note that in 350 days of 2014, 42 days (12%) were above average and 308 days (88%) were under the average.
In the 288 days of 2015 (until 15 October 2015), the trend changed, with 182 days (63.2%) above the long-term
average (Table 1).

To investigate the possibility of a repeating pattern or periodic signal in the record, the time series with the
number of sessions were analysed using the autocorrelation function (ACF). The ACF measures the degree of
correlation between a signal and the signal itself shifted by a given lag, and is defined as:

$$ACF = \frac{1}{n\sigma^2}\sum_1^{n-k}(X_i - \bar{x})(X_{i+k} - \bar{x}) \qquad \text{eq. (1)}$$

where $k$ is the lag (a day in this case), $n$ is the length of the time series (607 days), $\sigma$ is the standard deviation
of the values (i.e., the standard deviation of the number of sessions), $\bar{x}$ is the average of the values (i.e., the
average of the number of sessions), and $X_i$ is a given value of the time series (the value of the number of
sessions of the day $i$). Due to the evident increasing trend (non-stationary) in the average number of sessions
during the observation period (dashed line in Fig. 4), data have been detrended. The trend has been defined
fitting a curved line (Fig. 5a) obtained applying a kernel smoother based on a normal weight function in a
bandwidth of 100 days. Figure 5b shows the coefficients (ACF) calculated per different lag times. The
autocorrelation value varies between 1 and -1, and the area between the blue dashed lines represents non-
significant autocorrelation values. The analysis revealed that the value of ACF decreases when the lag $k$ (days)
increases, and that a marginally significant value of autocorrelation can be observed only for a lag of seven days (a week). However, because the correlation value is not significant at 14 or 21 days, we conclude that the time
series of the number of sessions of the POLARIS website does not show evidence of a periodic pattern. The
same analysis was performed detrending the data fitting a linear interpolation (dashed line in Fig. 4). Again, the
analysis did not reveal a periodic trend.

To gain a better understanding of the temporal distribution of the user access, and to identify peak values, we
used the daily number of users and pageviews obtained from Google Analytics. We then related the peak values
to several factors, including (i) the occurrence of harmful geo-hydrological events, (ii) the daily early warnings
from the Italian National Department of Civil Protection, (iii) the publication of new content in the web site,
(iv) the publication of press releases that used our data, and (v) the promotion of the website through media.

Figure 6 shows the daily user statistics (Fig. 6a), and a comparison between users and number of pageviews
(Fig. 6b), for the 21-month period of website publication, with icons located to identify 
[revised manuscript text omitted]

**7    Discussion**

In Polaris we mean risk communication as a two-way exchange of related information and knowledge on natural hazards and associated risk for the population. The Blog section of the website is mainly encouraging bottom-up feedback through visitors' s comments. The link to Facebook stimulates more feedback from citizens who upload pictures and make post on Facebook. This means that participation, whose central elements are influence, interaction and information exchange (Bostenaru, 2004), is mainly facilitated by the link with Facebook. However, the website Blog section remains less active than we expected, for at least two reasons: first, in Italy, the perception of geo-hydrological hazards is still very weak, people show less interested toward these geo-hydrological events than to other natural hazards such as, seismic risk (Salvati et al., 2014). Second, people do not know how a geo-hydrological event can hit them. People are interested to actively participate through the blog section mainly when a particularly disastrous event is occurring, and in such a case, by simply uploading videos and pictures rather than asking for explanation or advices. This means that, despite many institutions are making efforts to increase the public understanding of geo-hydrological risk through nationwide awareness campaigns (e.g. I do not risk, http://iononrischio.protezionecivile.it/), people still ignore how a large part of the Italian territory suffers of geo-hydrological risk. Such an underestimation of the possible risks, the high confidence in the local administrators towards which citizens delegate their personal safeness are all factors that impede an effective risk communication.

It is important to highlights that Polaris offers a knowledge-oriented risk communication which tends to operate continuously and does not regard the warning messages released in the event of a disaster. The communication efforts seeks to change the people's attitudes to the geo-hydrological hazard that they may have encountered giving many examples of what had happened before. People will not react to risk warnings if foregoing communication has not motivated and prepared them.

For this purpose, we are going to evolve the Blog section of Polaris which is the most relevant for stimulating public participation at any moment. In particular, we plan to integrate other relevant social media, such as

Instagram and Pinterest, stimulating the sharing of images and videos and the associated tags and comments.

For encourage more resilient behaviours during the occurrences of hazardous events, we would stimulate the usage of video through the YouTube and Vimeo channels that we can comment for feedback and/or advices.

Finally, we are going to create new synergies with the "I do not risk" campaign and website of the Italian

Department of Civil Protection, which will increase traffic, information exchange and, as such, strengthen the risk perception by the Italian population.

## 8   Concluding remarks

The analysis we conducted in the 21 months after publication of the POLARIS website allowed the following considerations. The geographical distribution of people interested in the published topics is widespread throughout Italy, with a few geographical areas in which sessions are highly concentrated. After the home page, the most viewed website section is the Report, followed by the Focus and Events sections. In a period shorter than two years, the number of sessions has generally increased; however, we observed that, in 2015, the most significant positive step occurred. The analysis of the time series, performed to identify possible periodical signals in the daily distribution of sessions, did not highlight any relevant information.

Monitoring the access of users to the POLARIS website and the number of pageviews during its publication period from 16 January 2014 to15 October 2015, we noticed that, frequently, the peak values correspond to the occurrence of particularly damaging geo-hydrological events. However, inspection of the daily statistics available for CNR IRPI Facebook demonstrated that a correspondence exists between the extreme rainfall events and the number of people who have viewed the content Facebook page. This finding was expected because CNR IRPI Facebook page's objective is to capture the attention of the public at large by proposing content that satisfies their curiosity and their immediate interest during extreme events, which increases the number of followers. Because the Facebook page is linked to POLARIS, an increase in Facebook followers can trigger a gradual increase in the number of people interested in more structured and specialized content and data
on geo-hydrological topics such as those published on POLARIS. Similarly, the specificity, scientifically based,
of the POLARIS content, which is focused on geo-hydrological hazard and risk, became a source of information
for journalists and media operators. The growth of user access when media operators publicized the website,
suggested that we enhance our collaboration with scientific journalists by linking traditional (e.g., television)
and social media to further enlarge the awareness of the website, and to better explain to users how to exploit
the website information.

The POLARIS initiative demonstrates how the scientific community can implement different communication
strategies to enhance an effective process that helps different audiences to understand (i) how risks associated
with geo-hydrological hazards are estimated and (ii) how risks can be reduced by increasing knowledge to the
population.

**Acknowledgments**

We thank Salvatore Buda and Vito Lo Re for the website design and info-graphic development, and Mauro
Rossi for making available the non-exceedance probability cumulative rainfall maps' rainfall data series. The
study was partially financed by the Italian National Department of Civil Protection (DPC). CB was supported
by a grant of the DPC.

**Figure captions**

**Figure 1.** The POLARIS communication flow.

**Figure 2.** The POLARIS Home Page (http://polaris.irpi.cnr.it). Violet boxes show English translation of original Italian text.

**Figure 3.** General statistics from Google Analytics for the 638-day period from 16 January 2014 to 15 October 2015. (a) map showing the geographical distribution of the sessions in Italy. (b) Pie chart shows number of pageviews for different sections of the website.

**Figure 4.** Daily average access number to the POLARIS website in the 638-day period from 16 January 2014 to 15 October 2015.

**Figure 5.** (a) Plot shows the original data (points) and the line (violet line) describing its trend. (b) Chart shows Autocorrelation Coefficient Function (ACF) calculated using the time series of the number of sessions of the POLARIS website.

**Figure 6.** (a) Daily number of users of the POLARIS web site in the 638-day period from 16 January 2014 to 15 October 2015. (b) Daily number of pageviews (violet line) and users (blue line) in the same period.

**Figure 7.** Number of unique Facebook page users. Days with extreme rainfall conditions are marked by blue dots, days with the major geo-hydrological events are marked by orange diamonds, and days with severe warning declarations are marked by red dots.

|  | Statistics | Number |
|---|---|---|
| Sessions | Total | 17,159 |
| | Daily average | 26.9 |
| | Average duration | 00:02:38 |
| | Days above average (2014) | 42 (12%) |
| | Days above average (2015) | 182 (63.2%) |
| Users | Total | 11,529 |
| | Daily average | 23.3 |
| | Days above average (2014) | 37 (10.6%) |
| | Days above average (2015) | 180 (62.5%) |
| Pageviews | Total | 44,032 |
| | Daily average | 69 |
| | Average per session | 2.6 |
| | Days above average (2014) | 68 (19.4%) |
| | Days above average (2015) | 165 (57.3%) |
| | Home page | 14,284 |
| | Report section | 5976 |
| | Focus section | 5509 |
| | Significant Event section | 5489 |
| | Blog section | 2550 |
| | Alert Zones section | 2108 |
| | Are You Prepared? section | 1894 |
| Referrals | Total from Social Network | 2394 |
| | Facebook | 1917 (80%) |
| | Twitter | 430 (18%) |
| | Other Social Networks | 47 (2%) |

**Table 1**: **POLARIS website general statistic for sessions, users, pageviews, and referrals from social networks,**
**calculated using Google Analytics data.**

[Figure]

**Figure 1: The POLARIS communication flow.**

[Figure]

**Figure 2: The POLARIS Home Page (http://polaris.irpi.cnr.it). Violet boxes show English translation of original**
**Italian text.**

[Figure]

**Figure 3: General statistics from Google Analytics for the 638-day period from 16 January 2014 to 15 October 2015.**
**(a) map showing the geographical distribution of the sessions in Italy. (b) Pie chart shows number of pageviews for**
**different sections of the website.**

[Figure]

**Figure 4: Daily average access number to the POLARIS website in the 638-day period from 16 January 2014 to 15**
**October 2015.**

[Figure]

Figure 5: (a) Plot shows the original data (points) and the line (violet line) describing its trend. (b) Chart shows
Autocorrelation Coefficient Function (ACF) calculated using the time series of the number of sessions of the
POLARIS website.

[Figure]

**Figure 6**: **(a) Daily number of users of the POLARIS web site in the 638-day period from 16 January 2014 to 15**
**October 2015. (b) Daily number of pageviews (violet line) and users (blue line) in the same period.**

[Figure]

**Figure 7**: **Number of unique Facebook page users. Days with extreme rainfall conditions are marked by blue dots,**

**days with the major geo-hydrological events are marked by orange diamonds, and days with severe warning**

**declarations are marked by red dots.**

---

## Author Comment (AC4) · 27 Apr 2016

Please find enclosed the revised version of the manuscript

Please also note the supplement to this comment:
http://www.nat-hazards-earth-syst-sci-discuss.net/nhess-2015-354/nhess-2015-354-AC4-supplement.pdf
* * *